# Cultural adaptation of a psychosocial screening tool for adolescents living with HIV/AIDS attending antiretroviral therapy program in Malawi

Esther C. Kip[1*], Mina C. Hosseinipour[1,2], Kazione Kulisewa[3], Brian W. Pence[4], Michael Udedi[5], Bradley N. Gaynes[4], Vivian F. Go[6]

1 University of North Carolina Project, Lilongwe, Malawi, 2 Department of Medicine, University of North Carolina School of Medicine at Chapel Hill School of Medicine, Chapel Hill, North Carolina, United States of America, 3 Psychiatry Department, Kamuzu University of Health Sciences, Blantyre, Malawi, 4 Department of Epidemiology, Gillings School of Global Public Health, University of North Carolina at Chapel Hill, Chapel Hill, North Carolina, United States of America, 5 NCDs and Mental Health Division, Ministry of Health, Lilongwe, Malawi, 6 Department of Health Behavior, Gillings School of Global Public Health, University of North Carolina at Chapel Hill, Chapel Hill, North Carolina, United States of America

* esther_kip@yahoo.com

## Abstract

### Background

Adolescents living with HIV (ALHIV) face a higher risk of mental health problems than adolescents without HIV yet culturally appropriate mental health screening tools are lacking in settings like Malawi. This study aimed to culturally adapt the HEADSS (Home, Education, Activities, Drugs, Sexuality, Suicide/Depression) psychosocial screening tool for the Malawian context, as the original was previously found to be inappropriate.

### Methods

The study was conducted between December 2021 and May 2022. We employed an adapted Heuristic Framework for cultural adaptation translations. Data was collected with Mental Health experts (n = 4), focus group discussions with ALHIV (n = 20), in-depth interviews with health care providers (HCPs) (n = 6) and key informants (n = 4). An iterative process of piloting and feedback guided the adaptation.

### Results

The adaptation addressed conceptually difficult, unacceptable, or stigmatizing items. HCPs reported that the adapted HEADSS tool may effectively guide the examination of ALHIV challenges and simplify the identification of high-risk behaviors (e.g., early sexual debut, self-harm and substance abuse). The tool's language was culturally

**Data availability statement:** All relevant data are within the manuscript.

**Funding:** The study was funded by a consortium between the North Carolina at Chapel Hill at Chapel Hill, Johns Hopkins University, Morehouse School of Medicine, and Tulane University (UJMT) with support from the Fogarty International Centre of the National Institutes of Health under award number NIH/FIC grant #D43 TW009340. ECK was awarded a post-doctoral fellowship from UJMT. The funders had no role in study design, data collection and analysis, decision to publish, or preparation of the manuscript. The content is solely the responsibility of the authors and does not necessarily represent the official views of National Institute of Health.

**Competing interests:** The authors have declared that no competing interests exist.

accepted by ALHIV, as the screening questions were available in both the local language and English, and accurately reflected their daily challenges.

## Conclusions

Developing easy-to-use, comprehensible, and locally appropriate mental health screening tools is crucial for detecting high-risk behaviors and psychosocial issues among ALHIV. To effectively meet ALHIVs' needs within HIV services, mental health interventions are essential for improving adherence to antiretroviral therapy (ART). Therefore, training HCPs to address sensitive risk issues during routine care is highly recommended.

## Introduction

Globally, mental health disorders are a major cause of disability among children and adolescents. They are associated with substance abuse, early sexual debut, HIV transmission risk behavior, and elevated suicide risk in adolescents. Mental health disorders are a particularly critical and neglected global health challenge for adolescents living with HIV. Further, these disorders undermine adherence to antiretroviral treatment, leading to poorer health outcomes among adolescents living with HIV (ALHIV) [1,2]. The prevalence of mental and behavioral health issues among ALHIV may not be well understood or addressed as the world scales up HIV prevention and treatment for adolescents [2].

The lack of culturally appropriate mental health assessment instruments is a major barrier to screening and evaluating efficacy of interventions [3–6]. Studies have shown that rigorous cultural adaptation and clinical validation procedures can ensure that assessment instruments are locally appropriate and valid [3–5,7–14]. Cultural adaptation is "the systematic modification of an evidence-based treatment (EBT) or intervention protocol to consider language, culture, and context in such a way that is compatible with the client's cultural patterns, meanings, and values" [3,5,7,12,14–16]. Cultural adaptation is warranted when an intervention developed for one cultural group will be implemented within a different cultural group [12, 15, 17, 18]. Adapting interventions to better suit a specific cultural group can lead to improved treatment outcomes. This is because cultural adaptations help to increase engagement and relevance, making the intervention more likely to resonate with individuals and encourage adherence to the treatment plan [4, 5, 19–21]. Culturally adapted interventions have the potential to improve both client engagement in treatment and outcomes and might be indicated when rates fall below what could be expected based on previous evidence [5,11,12,15,10]

The Home, Education, Activities, Drugs, Sexuality, Suicide/Depression (HEADSS) is a psychosocial screening designed to improve identification of psychosocial well-being and high-risk behaviors in adolescents [22–32]. It has been validated in outpatient departments in the US and found to accurately screen for mental health problems in young people with reported sensitivity of 82% and a specificity of 87%

in predicting psychiatric consult and admission to in-patient psychiatry [22–24,29,31–33]. It has been effectively implemented in emergency departments in the US and has also been used extensively in sub-Saharan Africa with pediatric hospital populations and adolescents living with HIV [24,30–35].

The HEADSS is a practical, time-tested strategy that physicians can use for psychosocial review of systems to evaluate how their teenaged patients are coping with the pressures of daily living including family, peers, school, culture and their inner world. Furthermore, this screening tool is used all over the world to screen and identify high risk behaviors in adolescents. It also helps in guiding health counselling, including commending and building on strengths, exploring options, planning actions, providing information, and identifying need for intervention and referral [26,27,34,36–40]. In Malawi, HEADSS screening holds promise for effective use at adolescent ART program [22] but our work suggests cultural modifications are required. Building upon our prior formative research which identified barriers and facilitators to implementing the HEADSS ALHIV in Malawi [22], this study adapted the tool between December 2021 and May 2022 to enhance its cultural appropriateness for the Malawian context. The formative findings indicated that healthcare providers perceived the original tool as too lengthy, time-consuming, and lacking cultural relevance/appropriateness. This adaptation aimed to improve the expansion of screening and identification of mental health issues among ALHIV in ART programs, with the ultimate goal of facilitating more holistic care for ALHIV. Therefore, our primary aim was to use a rigorous cultural adaptation process to develop the HEADSS tool for appropriate, understandable, and ease -of-use in adolescent ART services

## Materials and methods

### Study design

We conducted this cross-sectional qualitative study between December 2021 and May 2022.

### Study sites

The study took place in two ART program health facilities in Zomba District at Zomba Central Hospital (ZCH) and Likangala health center. These two sites were selected because they implement ALHIV activities and serve a high number of active ALHIV. We chose an urban site (Zomba Central Hospital) and a health center (Likangala) to increase the transferability of our study findings. However, funding limitations necessitated the selection of only two sites, which may impact the generalizability of the culturally adapted tool.

### Study Population

This study engaged both males and female ALHIV. They were included in the study if they were: [1] aged between 12 and 18 years; [2] receiving HIV treatment and care services from the selected ART clinics; [3] able to communicate in the local language (Chichewa); [4] allowed by a consenting parent or guardian to participate; [5] able to give assent and present themselves on the day of the interview. In addition, included HCPs working with ALHIV, key informants (KIs) at District Health Office (DHO), Zomba Mental Hospital and other psychosocial counsellors working for non-governmental organizations such as Elizabeth Glaser Pediatric AIDS Foundation (EGPAF).

### Sampling strategy and sample size

A purposive sampling approach was used to select the participants. A total of 34 participants were included in this current study. These comprised of Mental Health experts (n = 4) and we selected and interviewed ALHIV belonging to the age group 12–14 and to the age group 15–18 (during FGDs). We made every effort to ensure that our sample was gender balanced, totaling to 20 in the 2 sites. In addition, we engaged 4 KIs (Psychosocial Counsellor, District Health Promotion Officer, Psychology Educator and Zomba Mental Hospital Nursing Director), 6 HCPs (Table 2). We utilized purposive sampling to select participants whose specific characteristics directly aligned with the study's aims and who were most likely

to contribute valuable data. This approach is particularly effective when the research question necessitates a specific type of participant, such as experts, individuals with unique experiences, or those with particular demographic characteristics [41–43]. In this study, our selection included professionals with considerable experience and expertise in mental health and working with ALHIV within ART programs. We also sampled ALHIV in ART programs, as their direct experiences with psychosocial issues and characteristics were highly relevant to the research topic. Investigators set initial participants' sample size to interview, with the ultimate goal of achieving theoretical saturation defined as the point where no new information emerged from the study participants [44,45]. This methodological alignment inherently supports the iterative and exploratory nature of cultural adaptation, where initial insights from a selected group of participants guide subsequent refinements rather than relying on a single, comprehensive data collection effort [41,46,47]. This adaptive research design allows the sampling strategy to evolve as the understanding of the cultural context deepens. Furthermore, the sample size was constrained by limited resources.

## Data collection

Data collection was coordinated by the Principal Researcher (PR), who worked with one trained Research Assistant (RA) in social science and qualitative data collection. The RA had been trained on the study protocol, interviewing skills, and research ethics including specific issues related to research with human participants and ALHIV. Before the focus group discussions (FGDs) (S1S2 Texts in S1 File) English and Chichewa data collection tools, the interviewers clearly described the study goals and objectives and obtained informed assent or consent forms from each research participant (and from a parent or guardian for adolescents below the age of 18). They also explained how the data would be used and the procedures in place to protect the anonymity and confidentiality of informants. We conducted all FGDs in the local language (Chichewa). Some HCPs and KIs preferred to be interviewed in English, while others preferred or were comfortable to communicate in Chichewa. All interviews were digitally recorded. The IDIs with HCPs and KIs last for around 45 minutes while the discussions lasted up to one hour. Additionally, we took notes during the FGDs, which complemented the audio recordings during the transcription process.

## Ethical considerations

We conducted this study in keeping with guidelines related to research involving human subjects. All methods were performed in accordance with the relevant guidelines and regulations. The study protocol and tools were approved by the College of Medicine Research and Ethics Committee (COMREC Ref No.**P.05/21/3328**). The research team received training regarding child protection and were asked to sign a Code of Conduct on Child Protection. The study was conducted in accordance with provisions of the study protocol, the Declaration of Helsinki (October 2013) [48], the WHO Handbook for Good Clinical Research Practice (July 2005) [49] and privacy and confidentiality were guaranteed consistent with guidelines for research involving young people. At the eligibility screening stage, verbal assent from ALHIV and verbal consent from a parent or guardian was obtained. This approach is in line with the National Commission for Science and Technology sections 18 and 48 of the Science and Technology Act No.16 of 2003 for Malawi [50]. After verbal consent, both parents or guardians and those participants above 18 years had to sign the written consent forms and those below 18 years had to sign their written assent forms before contributing to the group setting. In this case participants had to identify themselves during the consenting process but not during the discussion.

Therefore, informed written consent was obtained from all participants and from their parent and/or legal guardian for participants below 18 years age. In addition, participants were informed that those who might feel uncomfortable while relating their experience of living with HIV, in which case interviewers will stop recording and stop the interview depending on the wishes of the research participants. Furthermore, they were explained that the research team would refer to specific services as needed in case of any emotional distress. The FGDs were conducted without their parents/guardians. To maintain confidentiality, we attribute quotations with only participant's sex and from which age groups.

## Adaptation procedures

This study used an adapted Heuristic Framework [51–53] for the cultural adaptation of interventions because it provides flexibility in the dimensions for adaptation and clear steps for structuring the cultural adaptation process [51–53]. Literature shows that there is growing consensus about the stages investigators might take to design adaptations to both engagement and intervention procedures. Lau's examples of adaptation strategies, other literature with adaptations suggest a possible sequence of adaptation stages [51,53]. The framework includes four stages that include a range of information gathering, preliminary adaptation design, preliminary adaptation tests and adaptation refinement [51–53]. This is a framework for guiding research that evaluates the cultural equivalence of evidence-based treatments (EBTs), and for practices in developing adaptations [51–53].

We conducted HEADSS screening tool adaptation in four stages. We applied an established, systematic process for adapting mental health screening instruments for children and adolescents. Recommendations from the previous formative research included the need for a culturally appropriate HEADSS tool for Malawi setting, culturally sensitive questions should be excluded (for example questions about sex orientation); that the screening tool should be in a flipchart format with some graphics/pictures; and it should be in both English and Chichewa (official languages in Malawi). These were integrated into the preliminary modifications of the adapted HEADSS screening tool.

**Stage 1**: In the initial stage we engaged a multi-disciplinary team of local Malawian mental health experts comprising of clinical psychologist, psychosocial counsellor, mental health nurse and teen club coordinator who is also a psychiatric clinical officer to give them an orientation of the adapted HEADSS. Two were from the Northern region where Tumbuka is widely spoken, one from central where Chichewa is widely spoken and one from south but from a Yao speaking territory. They all had to translate the content and back translated. This was done to ensure accuracy and cultural sensitivity. The changes were approved via consensus. The draft consensus translation was then back translated into English and Chichewa by one independent bilingual Malawian with a background education in Social Sciences to check the original language for discrepancies and ensure the meaning is sustained.

**Stage 2**: We conducted preliminary HEADSS tool content adaptation with this team of mental health experts. They reviewed the content in the adapted HEADSS for cultural and linguistic modifications and translated independently each item into the local language (Chichewa) and back translated with a focus on comprehensibility (retaining original semantic equivalence), appropriateness (fit, relevance, compatibility with cultural context), and a specific focus on ease-of-understanding. The changes were approved via consensus. The draft consensus translation was then back translated into English by 1 independent bilingual Malawian. Further modifications were made based on the back-translations.

**Stage 3** We conducted preliminary testing of the adapted tool. We conducted IDIs with KIs and HCPs, and FGDs with the ALHIV to solicit their views regarding comprehensibility, acceptability and relevance of the adapted HEADSS.

**Stage 4**: We finalized the HEADSS tool after incorporating feedback from additional meetings with IDI and FGD participants. Recommendations for further adaptation or refinement were documented.

## Data *analysis*

A reflexive thematic analysis technique was employed to analyze data [54–57]. Reflexive thematic analysis is an easily accessible and theoretically flexible interpretative approach to qualitative data analysis that facilitates the identification and analysis of patterns or themes in each data set [54–57]. Prior to familiarizing with the dataset, the primary researcher engaged in the iterative process of reflexivity [54,55]. The first step in the analysis was to read repeatedly through all the transcripts and took notes to obtain an overall understanding of the data in order to gain an in-depth understanding of the context, concepts, codes, and potential themes [58–60]. Data were transcribed verbatim, and transcripts were entered into NVivo 13 QSR International so that emerging themes could be identified, inspired by a deductive directed approach [58–60]. As part of code development process, we identified aspects of data that were interesting and could be useful

in developing themes for FGDs. Codes that appeared to be most relevant to the research question had been organized into meaningful themes [54,58,59]. Consequently, codes that were prevalent throughout the entire dataset were subsequently informative in the development of our themes. We then reviewed and analyzed coded data to generate themes and sub-themes. Finally, we developed a thematic framework, and each theme was defined. Direct quotes were obtained and used to explain and describe themes and sub-themes and to ensure that the results accurately conveyed the participants main points. These codes were refined through inductive analysis and results were organized according to identified key themes from the data. Analysis (thematic coding, generating deductive codes based on interview guide and adding inductive codes iteratively based on emergent themes) [54,58–60] was primarily conducted by primary researcher (ECK). Therefore, the results of the analysis represent author one's interpretations of the data. Furthermore, we applied the process of information triangulation through using different data sources such as interview transcripts to see if they support similar themes and also by conducting iterative interviews HCPs and KIs and FGDs with ALHIV. Additionally, we took notes during the FGDs, which complemented the audio recordings during the transcription process. Also conducting a document review provided us with relevant concepts and theories to engage with the study participants.

### Rigor and trustworthiness

Rigor and trustworthiness refer to the extent of confidence qualitative researchers have in their data [61,62]. This is assessed using criteria of credibility, transferability, dependability and conformability [61,62]. Accordingly, we have provided a rich description of the setting and context where we conducted the study to make our results transferable to other areas. Furthermore, increase the transparency of the interpretation, coding categories are illustrated with direct quotations in the presentation of the results.

### Reflexivity/positionality of the researchers

The researchers in this study have strong educational background in public health, social sciences, mental health and qualitative methods, and many years of qualitative research experience in a range of public health issues, including HIV, health and mental health care services. The primary researcher (ECK) is familiar with HIV, ALHIV context, through prior knowledge and interactions with the study population during the previous two studies with HCPs in this context and worked with organizations providing HIV care. Given the strong educational background and research experience of the researchers in the current study, it is believed that the research questions drove the methodology and methods employed to answer the research questions. ECK conducted data analysis and this analysis reflected on her beliefs around the subject before and during the analysis.

## Results

The findings are discussed along the themes and the sub-categories that were derived from the data. Appropriate direct quotes have been used where relevant to clarify the results and literature is provided to support the reported findings, where appropriate.

The participants were interviewed regarding comprehensibility, acceptability, relevance and overall impression of the adapted HEADSS.

### Demographics

We engaged mental health experts (n = 4), focus group discussions with ALHIV (n = 20), in-depth interviews with HCPs (n = 6), KIs (n = 4). Thus, a total of 34 participated in this study are shown in Tables 1 and 2. We selected different ALHIV belonging to the age group 12–14 and to the age group 15–18 (Table 1) with the median age of 15 years. Each of the groups were balanced by gender.

**Table 1. Demographic characteristics of adolescents living with HIV.**

| Study Sites and age ranges | Sex | | Median age | Total Number |
|---|---|---|---|---|
| **Zomba Central Hospital** | Female | Male | | |
| 12–14 years | 3 | 2 | 13 | 5 |
| 15–18 years | 2 | 3 | 16 | 5 |
| Total | 5 | 5 | | 10 |
| **Likangala Health Center** | | | | |
| 12–14 years | 2 | 3 | 13 | 5 |
| 15–18 years | 3 | 2 | 17 | 5 |
| Total | 5 | 5 | | 10 |
| Grand total | 10 | 10 | | 20 |

**Table 2. Demographic characteristics of mental health experts, health care providers and key informants.**

| Positions | Sex | | Age | Education | Total Number |
|---|---|---|---|---|---|
| **Mental Health experts** | Female | Male | | | |
| Clinical psychologist | 1 | | 42 | Tertiary | 1 |
| Psychosocial counsellor | 1 | | 32 | Tertiary | 1 |
| Mental health nurse | | 1 | 39 | Tertiary | 1 |
| Team club coordinator/psychiatric clinical officer | | 1 | 36 | Tertiary | 1 |
| Total | 2 | 2 | | | 4 |
| **Health Care Providers** | | | | | |
| Clinical officer | | 1 | | Tertiary | 1 |
| Mental health nurse/Adolescent focal person | 1 | | | Tertiary | 1 |
| Psychosocial counsellor | 1 | | | Tertiary | 1 |
| Team Club Coordinator | | 1 | | Secondary | 1 |
| Expert Client | 1 | | | Secondary | 1 |
| Adherence Support Officer | | 1 | | Secondary | 1 |
| Total | 3 | 3 | | | 6 |
| **Key Informants** | | | | | |
| Mental Health Nursing Director | 1 | | | Tertiary | 1 |
| District Health Promotion Officer | | 1 | | Tertiary | 1 |
| Psychology Educator | 1 | | | Tertiary | 1 |
| Psychosocial Counselor (EGPAF) | | 1 | | Tertiary | 1 |
| Total | 2 | 2 | | | 4 |
| | | | | | 14 |

[1] EGPAF: Elizabeth Glaser Pediatric AIDS Foundation

## Adaptation results

Several types of modifications were made from original HEADSS tool (S3–S9 Texts in S1 File) to address problems with comprehensibility, acceptability and its relevance including completeness.

## Comprehensibility and relevance of the adapted tool

In order to improve comprehensibility, several changes to HEADSS tool items were done. During the meeting with a team of mental health experts they commented that the adapted HEADSS looked better than the original one, however they

made further modifications in the adapted HEADSS by changing most of the words. The original and first version of the adapted HEADSS was only in English, and they translated it into Chichewa. Additional adaptations included adolescents being addressed in Chichewa by the plural 'mu' rather than singular 'u' in line with cultural norms of being respectful to the patient.

During the adaptation process, several questions were removed from the original HEADSS tool due to their unsuitability or lack of clarity for the Malawian context. Specifically, questions pertaining to sexual orientation, such as "have you ever had sex with men? Women? Both?", "do you think you might be lesbian, gay, or bisexual?", and "do you think you need to have sex to find out if you're lesbian, gay, or bisexual?", were omitted. Additionally, certain questions on suicidal ideation, including "how do you feel today, on a scale of 0 - 10 (0 = very sad, 10 = very happy)?", "have you ever felt less than a 5?", "how long did that feeling last?", "what made you feel that way? does thinking you may be lesbian, gay, or bisexual make you feel that way?", and "did you ever think that life isn't worth living, or hope that when you go to sleep you won't wake up?", were also excluded based on participant feedback. The KIs and HCPs indicated that the adapted HEADSS will guide to better systematic counselling, better client provider relationship, might improve ALHIV quality of care, will give ALHIV some hope for the future, good for holistic psychological assessment of ALHIV, they felt that the tool is clear and user-friendly as one highlighted " *it looks holistic and might be a good guiding tool to improve the welfare of the adolescents"*. Among the HCPs cadres, there are Expert Clients. Expert Clients are trained HIV Positive individual who workers in health facilities in Malawi due to shortage of health care professionals. They act as lay health care workers, peer educators and supporters and they provide counselling, psychosocial support and assistance in accessing HIV care to children as well as pregnant women.

*It's a nice tool and it will improve the interaction especially between the youth (adolescents) who are the target group and the service providers. At least it is also giving some insights on how they think and how they perceive the world though I feel like some information needs to be added to probe more because this tool will be more like interaction in nature and there are other questions which are too short [Male Expert Client].*

*This tool is very important, it can help our adolescents and not just the adolescents but it can build a better relationship between a health care provider and the adolescents and also the parents and between the care providers and teachers as well as between the teachers and the parents. If health care providers have identified a problem which originates from school, then they can link up with the teachers and if the problem originates from home, then health care providers have to find a way of communicating with the parents to find a solution to end this problem. Thus, there will be good relationships, and this will help in improving the health outcomes of ALHIV and they will have a better future as well then, this adolescent will be a better reliable citizen of Malawi [Female key informant].*

*Among the 3 adolescents that I interviewed, I noticed that after they had seen the pictures or after going through a few sections, they were able to relate the pictures and the questions and they opened up; unlike before when they did not want to tell us more issues. Today, during the pilot testing of the tool, I encountered two adolescents who told me their experiences they are going through in their homes and at school. One indicated that at home his father is very violent and he abuses them all including his mother. He even thought of running away from home because of how his father abuses them all. The second adolescent said he got punished at school for something he did not do and he thought the teacher does not like him because of his HIV status. This affects him and he has once repeated a class [Teen Club Coordinator].*

*These questions in this tool are extremely important to ALHIV needs……With this tool, when we will be interviewing such adolescents, we will right away detect/identify a problem before it is too late and will be able to find a solution to such a challenge before it's too late. If one identifies that the problem is from school, then we would be able to find out if the problem originates from the teachers or her/his friends. If we leave the problem for a long time, ALHIV lose hope*

*and they start having thoughts that they are not worth living and be productive citizens of Malawi. So, this tool is very important to ALHIV because it can even encourage him as well [Clinical Officer].*

FGDS with ALHIV showed that they felt that the tool is very good because it encompasses all issues that affected their daily lives, and it can be used in Teen Club program because it can help identify those with some psychosocial challenges and would help the HCPs in counselling them.

*This tool is very good, and it can be used in Teen Club (TC) program because it can help the health workers identify those with some worries and they can be counselled. In addition, it can also help those who are being stigmatized and discriminated against because of their HIV status. When HCPs ask such questions, they can identify those ALHIV who are going through challenging situations and then they can counsel them (Female participant 12–14 yrs group).*

*When we are been screened, those questions on their own will help us change our behavior. We will be able to behave better. In this tool, there is that section on "suicide". If an adolescent has some ideas of committing suicide, when he or she is asked such questions, he or she can change his or her mind because the health workers can counsel him/her about the negative consequences of committing suicide and if that individual had such thoughts, he/she might change such thoughts [Female participant 15–18yrs group].*

*I say the questions in this tool are very important to ALHIV because if the HCPs use these questions to interview us and find out that we are going through a lot of problems, they will give us some counselling according to the challenges we are facing, because if that adolescent answers these questions, the HCP will know where to start then they will know how to help that individual by guiding and counselling her/him [Male participant, 12–14 years age group].*

## Acceptability of the language, mains domains, sub-domains and design

Regarding the language, all participants highlighted that the language used in the adapted tool was appropriate to Malawi setting since it was in the local language, Chichewa. Participants also indicated that the language utilized was culturally appropriate, respectful and had no insensitive or offensive words.

*….The two languages that have been used in this tool are very appropriate here in Malawi, especially in southern part. Both English and Chichewa will help us ask the questions properly and provide necessary assistance [Male Clinician].*

*Everywhere I've read, I didn't find any words that are so strong, sensitive and offensive. Most words used in this tool are common words the adolescents learn at school even about "Sexual and Reproductive Health" they learn at school…both language and wording are fine and appropriate to Malawi setting [District Health Promotion Officer].*

*The wording is acceptable because it is in Chichewa our local language in Malawi which is spoken from Nsanje to Chitipa (Meaning from Southern to Northern region). So, there is no one who can say s/he doesn't understand Chichewa because even in our primary schools they use Chichewa. It's like a national language [Mental Health Nursing Director].*

*The language is culturally accepted, because you have used the words that are straight forward and not ambiguous. There is nothing like offensive words ……"Otukwana" (obscene) [Psychosocial Counsellor].*

*For us, health care providers there is nothing like "kulawulana" (indecent) language. So, when I checked the HEADS tool, I did not see any offensive word which is not appropriate for a child. Each word that was used is acceptable in our Malawian setting [Adherence Support Officer].*

*The wording is fine. The HEADSS is not in Chinese, it is in English and Chichewa the most common languages in Malawi [Female participant 12–14 yrs age group].*

*The wording is acceptable because here in Malawi, we speak Chichewa, whether one is educated or not [Male participant 14–15 yrs old age group].*

*It is very acceptable because it's using day-to-day language. It is not irritating; Malawi culture is a bit different because we really respect ourselves. For example, if you are teaching a child, you know…..the private parts of the body…..you know, we don't mention, for example, we don't say, go and urinate, we say, go and pass water and so on. So, you have really respected the culture. It's very respectful. Yeah…….because as I have said, those other things that are hard to talk in Malawian setting for example about sex……so actually when we use this tool three or four times and we talk about those things, the adolescents will definitely accept it. For example, if we just do it today and we leave it, then it will be hard for them to understand what we are doing with this tool [Clinical Officer].*

Furthermore, all participants noted that all sections including the new sub domains were very essential.

*All sections touch the life of adolescents. There is a section that talks about home, school and food, as mentors we also teach them about nutrition as the tool has shown about the six food groups, disclosure as well which is already there in the tool. All sections will help us see how we can help, so all the areas are very necessary together they make one component, from home to school and then teen club…. because without using the tool it will not be possible for us to know some of the challenges the child is facing [District Health Promotion officer].*

*I find all sections useful, starting with home, children spend 80% of their time at their homes. If a child has psychological problems at home and if that child goes to school, s/he will still have problems. I spotted three sections, home, education and sexual reproductive health that I like most [Mental Health Expert].*

*According to my experience, issues on "Stigma and Discrimination" are very important issues to tackle because the adolescents are very affected. The adolescents have such a big burden that they are unable to express their issues [Adolescent Teen Club Coordinator].*

*………..Yeah, because you have used holistic approach. You have looked an at an adolescent in totality focusing on each section/area and looking at what issues these adolescents face in such areas [Mental Health Nurse].*

Additionally, ALHIV also expressed that the adapted tool's new sub-domains which were added in during the adaptation process under "Home and Environment domain" such as adherence to ART, Abuse, stigma and discrimination and food security sections were very useful.

*All the sections in the adapted tool are very acceptable because, all these are questions that touch our daily lives. So, there is no section that is not acceptable, all these sections are "bhooo bhoo" (meaning they are good and acceptable) …….all these are acceptable among us adolescents [Female participant, 14–18 years age group].*

*Yes, these questions in the HEADSS are important because this is what we encounter in our everyday lives. For example, "Abuse" which even causes some of the adolescents stop taking their medications because of the circumstances in their lives is very important section [Male participant, 15–18 years age group].*

There were mixed responses on sexual reproductive health, suicide, and depression sections from some of the ALHIV as well as the HCPs.

*I don't agree with the section on "Sexual and Reproductive Health". It is not useful to us. Because it does not help us in our daily lives [Male participant, 12–14 years age group].*

*…………..on Sexual and Reproductive Health section, there is a question which asks, 'how about you, have you ever been in sexual relationship and are you comfortable with it?' for some of 12-year old adolescents; they will feel shy to talk about sexuality. It will require us mentors (HCPs) to ask that question more carefully because otherwise their responses will be "no" if we ask them directly…if we do so parents may end up coming complaining that we have started teaching teens about family planning hence they may tell their child to stop attending teen club sessions [Teen Club Coordinator].*

*I find the "suicide section" not useful. Maybe if an adolescent is asked such a question on "suicide", it might bring in some ideas. It might be that that adolescent is going through some challenges in life and if s/he is asked questions on "suicide", he might just feel like committing suicide [Female participant, 12–14 years age group].*

*That picture on "Suicide" section is not good because sometimes, the children just hear about suicide or that someone has committed suicide but they have never seen how it is done. So that picture on suicide section might give some ideas to them because it doesn't mean that we will always be with that adolescent even if we counsel him/her here at the clinic. He will go out there and face challenges and the first picture s/he might have in mind it that "suicide" picture, so this picture is too graphic, it has to be replaced. In addition, some adolescents are being cared for by their guardians, their parents passed away because of suicide issues. This part which is asking if the adolescent have ever had suicidal thoughts, even though such questions will help us to know the number for instance…. I feel it's a strong content to discuss with adolescents about suicidal ideation [Expert Client].*

With regard to the tool's design, the participants commented that the adapted HEADSS was better because of the visual graphics as one highlighted: "*Yeah, I feel that the design is acceptable. We can relate it to other tools which we also use, like I said earlier on, …….it is not completely new…… We also have a discloser chart which looks like the HEADSS tool….it is familiar with other tools which we also use, so the design is just good [Psychosocial Counsellor].* Initially, the first set of pictures were cartoons and they were not accepted by majority of the participants, so we had to change the graphics several times and also there were some reservations on the picture in the suicide section which was eventually changed.

## Overall impression of the adapted HEADSS

Overall, the HEADSS tool was well accepted by both ALHIV, HCPs and KIs. It was clear that all participants felt that this HEADSS screening tool is needed for ALHIV. When it was piloted, some of the HCPs expressed that they were able to detect some high-risk behaviors such as non-adherence to ART, early sexual debut, substance abuse and suicidal ideation as some highlighted "*One adolescent said this experience of taking ARVs is very painful that when he is alone, he is gets stressed up and worried about his future. We went ahead to find a solution because I could see that there is some danger [Expert Client].* Some HCPs mentioned that it was an eye opener and this tool might help ALHIV improve quality of life as one highlighted……*You know, such issues are the ones little by little can influence these young people to have some suicidal thoughts/ideas. He thinks he cannot contribute much to their country because they feel hopeless. Without using HEADSS approach, we cannot identify such problems. It really shows that this tool is very essential to these ALHIV [Mental Health Nurse].*

## Discussion

This study's aim was to culturally adapt a multi-faceted HEADSS screening tool for identifying adverse psychosocial circumstances that place ALHIV at higher risk of common mental disorders, and risky behaviors such as early sexual debut substance abuse and non-adherence to ART among ALHIV in Malawi. We applied an established, systematic process for

 

adapting mental health screening instruments [3,5,9], which has been adapted for use with children and adolescents. This process ensures correct translation not only of language, but also of full equivalence of meaning and application of tools [3,14,4,20].

We have shown how systematic adaptation can be useful to improve acceptance, relevance, comprehensibility, and completeness of a screening tool. We were able to capture and cover the crucial elements to make the finished product in concurrence and sensitive to the context and experiences of the participants. In turn, we were able to make HEADSS culturally specific for this group, which is essential for mental health and HIV interventions among ALHIV. Our findings indicate that participants perceived the culturally adapted HEADSS to be relevant, acceptable/appropriate and comprehensible to Malawi setting. Thus, current results are in line with studies that have shown that rigorous cultural adaptation procedures can ensure that assessment instruments are locally appropriate and valid [3,5,6,8,9,15]. Some words that were identified as unacceptable or stigmatizing were removed and participants had to choose the appropriate words. Some domains or illustrations that were also deemed unacceptable to Malawi setting were modified since participant feedback guided iterative aspect cultural adaptations, which were again shown to participants for validation and improvement of screening tool items as mentioned earlier. Culturally adapted interventions have the potential to improve both client engagement in treatment and outcomes [3,14,18,63].

Participants indicated that this adapted HEADSS will facilitate in identifying psychosocial issues and high-risk behaviors among ALHIV. This finding also shown during pretesting of this adapted tool. Some ALHIV reported that they were being abused at home and school as well as they were engaged in substance abuse. However, this was a small number since we only pretested the tool on 20 ALHIV. HEADSS screening plays a role in the identification of psychosocial issues, such as depression and high-risk behaviors, among adolescents [27,28,31,32,35–38,64]. HEADSS has been applied effectively in some sub-Saharan African countries. Prior studies such as in Kenya, a study assessing 300 adolescents with the HEADSS identified 12 with suicidal ideation [32]. The researchers commented that this finding alone was a strong justification for routine use of HEADSS tool in assessing the ALHIV [32]. In Ghana HEADSS approach found a high psychosocial burden and a higher risk of mental health problems among ALHIV [22]. The factors identified with depression included unfavorable home situation, body image concerns, social isolation and legitimate HIV diagnosis –related fears such as stigma, dying young, and not marrying or having a sexual relationship [22]. In Tanzania, the National Adolescent Health and Development Strategy 2018–2022 recommended the HEADSS assessment for finding out issues among adolescents and providing necessary support which is usually not offered in facilities [22]. A recent study in South Africa (2024), showed that the three most-associated constructs from HEADSS were violence exposure, depression and sexual debut, were associated with increased ART non-adherence from 20.4 to 55.6% [64]. The three most-associated constructs medication side-effects, low social support and parents unaware of adolescent HIV status, were associated with increased ART non-adherence from 21.6 to 71.8% [64]. The researchers commented that these findings indicate valuable indicators for ART non-adherence [64].

Our current study found that both ALHIV and HCPs suggested removing questions about sexual orientation. This recommendation likely stems from the significant legal and societal challenges faced by sexual minorities in Malawi. The Penal Code in Malawi prohibits same-sex sexual activity, criminalizing "carnal knowledge against the order of nature" and "gross indecency" with a maximum penalty of 14 years imprisonment [58]. Sexual minorities, including men who have sex with men (MSM) and transgender individuals, face substantial barriers in the context of HIV/AIDS. The criminalization of same-sex sexual activity and deeply ingrained societal taboos foster a climate of fear, which in turn hinders their access to vital health services and increases their vulnerability to HIV and other STIs [58–61]. For those ALHIV who may identify themselves as lesbian, gay, bisexual, transgender, queer or questioning (LGBTQ+) may experience heightened stigma and discrimination from both their communities and HCPs, further impeding their access to essential sexual and reproductive health (SRH) services. However, removing questions about sexual orientation from surveys and censuses could severely hinder data collection on key populations. This lack of data could negatively impact policy development, resource

allocation, and research efforts focused on critical health, social, and economic outcomes [62]. The pervasive stigma and psychological distress among sexual minorities in Malawi significantly impact HIV transmission, treatment, and mental health [58–61,63–65]. This can lead to increased risky sexual behaviors, decreased treatment adherence, and higher rates of mental health disorders, thereby fueling the HIV epidemic. Specifically, psychological distress, including depression and anxiety, has been linked to increased engagement in risky sexual behaviors, such as unprotected sex, especially among those experiencing stigma [66–68]. Fear of judgment, discrimination, and potential legal repercussions actively discourage sexual minorities from seeking HIV testing, treatment, and other essential health services. Addressing stigma and psychological distress among sexual minorities in Malawi is not only a matter of human rights but also a crucial public health imperative. By promoting understanding, challenging discrimination, and ensuring access to essential services, Malawi can foster a more inclusive and equitable society for all, ultimately strengthening its HIV response.

Additionally, the HCPs, psychosocial counsellors and ALHIV were not keen on the questions around suicide. In Malawi, there is a common misconception or myth that if one asks someone who has poor mental health or adverse psychosocial circumstances whether they have suicidal thoughts, then that individual is suggesting or planting wrong thoughts to that person with poor mental health and that it might make it more likely that they might attempt suicide. However, this has not been proven by research in Malawi but such misconceptions have also been reported elsewhere [65,66] among patients, parents and caregivers, and providers that asking individuals about suicide could "put the idea in their head," and be harmful instead of helpful [66–68]. Contrary to these myths, multiple studies evaluated this iatrogenic risk and refuted this myth [65,66,69,70] These studies demonstrate that asking people directly about suicide rarely contributes to increased distress and does not cause someone to consider suicide [66,69,71,72]. Studies further suggest acknowledging and talking about suicide may in fact reduce, rather than increase suicidal ideation, and may lead to improvements in mental health in treatment-seeking populations [73–76]. A study in Ethiopia showed that prevalence of depressive symptoms among HIV-positive youth was 35.5% (95% CI:31.3, 39.6) [77]. In Uganda, the prevalence of suicidal ideation was 17%. The prevalence of suicidal ideation among ALWH was substantial. Children and adolescents with exposure to family or friend's death, those with higher depression scores, anxiety symptoms and rule breaking behavior were more likely to report suicidal ideation [78]. Suicidal behavior and HIV/AIDS are vital public health challenges especially in low and middle-income countries [77,79–81]. In Malawi, HCPs generally recognize the importance of screening for suicidal ideation, but face significant barriers to its implementation, particularly in primary care settings. While they acknowledge the potential benefits of early identification and intervention, practical challenges like limited resources, lack of training, and cultural context concerns hinder widespread adoption [22]. A major obstacle is the scarcity of mental health professionals and resources in Malawi. This includes a lack of trained personnel to conduct screenings and limited access to mental health services for those identified as at risk [82]. Another major challenge of successful integration of mental health into primary health care (PHC) is the lack of adequate knowledge, positive attitude, and skills for mental health service of HCPs participating in care and treatment of peoples at primary health care levels [24,83,84]. HCPs, especially those in primary care, need adequate training and ongoing support to effectively screen for suicidal ideation and manage individuals at risk. Training is successful at imparting knowledge, building skills, and molding the attitudes of trainees regarding suicide prevention [22, 85–89]. Including a suicide section in the HEADSS screener is very crucial for early identification and intervention in ALHIV at risk of suicidal ideation or behavior. HCPs might be able to screen for potential issues and address mental health concerns effectively and enabling a more comprehensive understanding of the ALHIV well-being and facilitating timely support.

This paper provides a unique opportunity to recognize that there is more to be done to improve the HIV and mental health outcomes among ALHIV in Malawi. Integration of mental health care into primary care is one of the most effective ways of reducing the substantial treatment gap for mental disorders which exists in most low- and middle-income countries [85,86]. Future studies are needed for integration of HEADSS screening in ALHIV care and also capacity building of HCPs in HEADSS screening as part of efforts to scale-up the mhGAP program nationally. These trainings should include processes for referral to

specific care providers for those individuals identified as potentially experiencing a mental disorder. Further adaptation of the HEADSS tool should include providing information on what causes suicidality and maybe training of the HEADSS tool should involve training on suicide so that providers and peers should have a degree of comfort in asking that particular domain. The findings of this study yielded evidence on both the scientific and practical, usability, acceptability, and feasibility of implementing HEADSS screening in the existing ART service delivery for ALHIV to address their critical needs. Currently a mixed-methods prospective quasi-experimental study employing a non-equivalent control group design and exploratory qualitative designs is being conducted to evaluate/validate this adapted HEADSS in clinical setting. Upskilling and capacity building of HCPs have been done to detection of risky behaviors and psychosocial issues among ALHIV in selected health facilities in Malawi.

## Strengths and Limitations

This study has a number of strengths as it was carried out in well-established ART programs for ALHIV in rural and urban settings. All ALHIV were on ARVs. It was conducted during the weekends so that adolescents in boarding schools had an equal opportunity to participate like those in day schools. To reduce gender bias a similar proportion of male and female adolescents were included in the focus group discussions from each site to ensure gender balance. The inclusion of different health care professions with some characteristics specific to the study aim, in light of their professions and experiences was also essential. However, despite the fact that this study produced valuable findings, the generalization of these findings is subject to a number of limitations. The sample (ALHIV) used has potential for selection bias as it was a sample chosen from ART program, which might not be generalizable since we would not be able to include those who did not attend this program. Additionally, purpose sampling method may only apply to this particular group studied and may not be representative of the larger cultural context or other subgroups within that culture. However, we included participants from various socioeconomic backgrounds, age groups, and other relevant demographics to ensure comprehensive feedback. We believe that adapting HEADSS culturally is crucial to identify at-risk ALHIV population in Malawi (and sub-Saharan Africa).

## Conclusions

Developing easy-to-use, comprehensible, and locally appropriate mental health screening tools is a vital first step in leveraging and detection of mental disorders among ALHIV. A culturally adapted screening tool, especially one that can be used by non-clinicians such as lay health workers, would improve the ability to address mental health needs of ALHIV in many primary care and social service settings where resources for professional mental health staff are limited. If HIV services are to effectively meet ALHIVs' needs, mental health interventions are needed to prevent and manage depression and improve adherence to ART. Upskilling and capacity building of HCPs can lead to better detection of risky behaviors and psychosocial issues among ALWH.

## Supporting information

**S1 File.** S1 Text. English Focus Group Discussion Guide. S2 Text. Chichewa Focus Group Discussion Guide. S3 Text. Original HEADSS tool. S4 Text. Participants HEADSS adaptation notes_v1. S5 Text. HEADSS adaptation v1. S6 Text. Participants HEADSS adaptation notes_ v2. S7 Text. HEADSS adaptation v2. S8 Text. HEADSS adaptation v3. S9 Text. HEADSS adaptation _v4_Final Version.
(ZIP)

## Acknowledgments

The authors would like to thank the participants for accepting to be interviewed and their time. Our deepest gratitude goes to the University of North Carolina, Kamuzu University of Health Sciences and Ministry of Health. The content of the article is solely the responsibility of the authors and does not necessarily represent the official views of the National Institutes of Health, University of North Carolina at Chapel Hill, Kamuzu University of Health Sciences in Malawi.

## Author contributions

**Conceptualization:** Esther C. Kip, Mina C. Hosseinipour, Kazione Kulisewa, Brian W. Pence, Bradley N. Gaynes, Vivian F. Go.

**Data curation:** Esther C. Kip, Mina C. Hosseinipour, Vivian F. Go.

**Formal analysis:** Esther C. Kip, Mina C. Hosseinipour, Vivian F. Go.

**Funding acquisition:** Mina C. Hosseinipour, Brian W. Pence, Bradley N. Gaynes, Vivian F. Go.

**Investigation:** Esther C. Kip, Mina C. Hosseinipour, Vivian F. Go.

**Methodology:** Esther C. Kip, Mina C. Hosseinipour, Vivian F. Go.

**Project administration:** Esther C. Kip.

**Resources:** Esther C. Kip, Mina C. Hosseinipour.

**Software:** Esther C. Kip.

**Supervision:** Esther C. Kip, Mina C. Hosseinipour.

**Validation:** Esther C. Kip, Mina C. Hosseinipour, Kazione Kulisewa, Brian W. Pence, Michael Udedi, Vivian F. Go.

**Visualization:** Esther C. Kip, Mina C. Hosseinipour, Michael Udedi, Vivian F. Go.

**Writing – original draft:** Esther C. Kip, Mina C. Hosseinipour, Vivian F. Go.

**Writing – review & editing:** Esther C. Kip, Mina C. Hosseinipour, Kazione Kulisewa, Brian W. Pence, Michael Udedi, Bradley N. Gaynes, Vivian F. Go.

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
