## [Decision Letter · Decision Letter 0]

11 Apr 2025

Cultural adaptation of a psychosocial screening tool for adolescents living with HIV/AIDS attending antiretroviral therapy program in Malawi

PLOS ONE

Dear Dr. Kip,

Thank you for submitting your manuscript to PLOS ONE. After careful consideration, we feel that it has merit but does not fully meet PLOS ONE’s publication criteria as it currently stands. Therefore, we invite you to submit a revised version of the manuscript that addresses the points raised during the review process.

We look forward to receiving your revised manuscript.

Kind regards,

Anthony A. Olashore, MBCHB, PhD.

Academic Editor

PLOS ONE

3. We note that your Data Availability Statement is currently as follows: All relevant data are within the manuscript and in Supporting Information files.

**Comments to the Author**

1. Is the manuscript technically sound, and do the data support the conclusions?

Reviewer #1: Partly

Reviewer #2: Yes

2. Has the statistical analysis been performed appropriately and rigorously?

Reviewer #1: Yes

Reviewer #2: Yes

3. Have the authors made all data underlying the findings in their manuscript fully available?

Reviewer #1: Yes

Reviewer #2: Yes

4. Is the manuscript presented in an intelligible fashion and written in standard English?

Reviewer #1: Yes

Reviewer #2: Yes

Reviewer #1: Title: Cultural adaptation of a psychosocial screening tool for adolescents living with HIV/AIDS attending antiretroviral therapy program in Malawi

Thank you for asking me to review the above-named manuscript. This manuscript describes a qualitative study that adapts the HEADSS psychosocial screening tool for adolescents living with HIV/AIDS in Malawi. The study is relevant, cultural adaptation of screening tools is important to ensure accurate detection of mental health conditions and design appropriate interventions. Find below my suggestion for improving the manuscript to make it suitable for publication.

Introduction

1. I would recommend that the authors talk more about the cultural modifications required. In the abstract, the authors stated, “We previously showed that the original HEADSS was not 23 culturally appropriate for the Malawian setting.” The authors need to provide more information about this, this would provide adequate rationale for the current study.

Methods

1. Study sites: could the authors provide a rationale for the selection of the 2 study sites? How does the selection impact the generalizability of the culturally adapted tool?

2. How many languages are spoken in Malawi? Are there different tribes? This would provide information about the utility of the adapted tool and external validity.

3. Adaptation procedures

a. Lines172 – 173: it would be helpful to state what the recommendations from the previous formative research were

4. Curious to learn more about how the mental health experts were selected. In addition, how did the mental health clinicians categorized under ‘healthcare providers’?

5. Were the mental health experts, healthcare providers, and key informants selected from the 2 ART centers too?

6. What language were the interviews conducted in? It was stated that the focus groups were conducted in the local language but not clear what language the interviews were conducted in

7. The process of thematic analysis could be further expanded. How many people participated in the coding process, was inter-coder reliability, member checking or triangulation used? how was consensus reached? It might be helpful to add a thematic map.

Results

1. It would be good if the author presented a table with the original HEADSS, the adapted version and the final version.

2. Spelling errors in the results section—specifically the quote from the participants.

3. Line 480—the authors should clarify ‘expert client’. Also repeated in lines 553 – 554.

Discussion

1. Lines 617 – 618: citation needed

2. Future studies should include validation of the adapted HEADSS. Psychometric properties such as face, concurrent and convergent validity and reliability are essential to show the appropriateness of a screening tool.

3. The authors should also touch on how the screening tool will be implemented and evaluated in clinical settings

4. The authors should comment on how the elimination of suicide screening will affect the validity of the tool and the impact on mental health outcome. Suicide is rising amongst adolescent population and patients with HIV/AIDS are particularly high risk.

Strengths

1. I wonder how the study setting provides strength for the study or how all the participants being on ARVs is a strength. Similarly, the authors should state why having participation of students in boarding school is a strength. What proportion of the participants are from boarding school?

Reviewer #2: This is a very important and rigorously done work. Just a few comments:

1.It would have helped to have a table with a few example questions of the adapted tool perhaps juxtaposed with the relevant changed questions of the HEADDS version used in your earlier work

2. Line 79: Although referenced it will help to briefly outline here the findings in your earleir work that suggested cultural modifications need to be done.

A few edits:

3. Line 435: , remove the word "to" before Malawi to make the sentence easier to understand

4. Line 556: add help before ALHIV

5. Line 593: the sentence "HEADSS screening has in the identification of psychosocial issues, such as depression and high-risk behaviors in adolescents (20,21,24,25,28–31,55)".is hanging and not clear.

6.Line 661-664: This last sentence under conclusion about training could be left out without loss to the paper as it does not tie in with the overall aim of the study and the discussion

**Do you want your identity to be public for this peer review?** For information about this choice, including consent withdrawal, please see our Privacy Policy

Reviewer #1: **Yes: ** Temitope Ogundare, MD, MPH

Reviewer #2: No

---

## [Author Response · Author response to Decision Letter 1]

19 May 2025

Authors 'Response to Reviewers

P. O Box 212

Zomba

Malawi

19th May, 2025

To:

Academic Editor

PLOS ONE

Dear Academic Editor Prof Olashore,

Ref: Re-submission ID PONE-D-24-58777: Cultural adaptation of a psychosocial screening tool for adolescents living with HIV/AIDS attending antiretroviral therapy program in Malawi

Authors:

Esther C. Kip (esther_kip@yahoo.com)

Mina C. Hosseinipour (mina_hosseinipour@med.unc.edu)

Kazione Kulisewa (kkulisewa@kuhes.ac.mw)

Brian W. Pence (bpence@unc.edu)

Michael Udedi (mphatsoudedi@yahoo.co.uk)

Bradley N. Gaynes (bgaynes@med.unc.edu)

Vivian F. Go (vgo@live.unc.edu)

Version 2: Date:19/05/2025

To the Academic Editor:

The authors would like to thank you and the reviewers for the valuable comments you have given us. Care has been taken to improve the work and address the reviewers’ concerns as per the specific comments below. We have used yellow highlighter to indicate the revised portions for the modifications in the manuscript. A point-by-point response detailing the changes we have made is indicated below. Overall, we are very grateful to the editors and reviewers for their positive and constructive suggestions. We feel that the quality of the revised manuscript has been improved significantly as a result. All comments in the manuscript file have been considered. We have followed each of the Editor’s recommendations for the revised version. We have used track changes for all the changes we have made.

REVIEWER 1 COMMENTS

Introduction

Reviewer comment: 1. I would recommend that the authors talk more about the cultural modifications required. In the abstract, the authors stated, “We previously showed that the original HEADSS was not 23 culturally appropriate for the Malawian setting.” The authors need to provide more information about this, this would provide adequate rationale for the current study.

Response: Thank you for your valuable feedback. In line with your comment, we have included more information about the cultural adaptation and the rationale as shown in see lines 70 – 81 p. 3 and 99 to 106 p.4 & 5).

Methods

Reviewer comment: 1. Study sites: could the authors provide a rationale for the selection of the 2 study sites? How does the selection impact the generalizability of the culturally adapted tool?

Response: Thank you for pointing this out. We have now included this sentence “These two sites were selected because they implement ALHIV activities and serve a high number of active ALHIV. We chose an urban site (Zomba Central Hospital) and a health center (Likangala) to increase the transferability of our study findings. However, funding limitations necessitated the selection of only two sites, which may impact the generalizability of the culturally adapted tool” as shown in lines 117 to 122 p.5.

Reviewer comment: 2. How many languages are spoken in Malawi? Are there different tribes? This would provide information about the utility of the adapted tool and external validity.

Response: Thank you for your inquiry. There are different languages spoken in Malawi and yes, there are different tribes. However, the most prominent are Chichewa, which is the official language along with English and Tumbuka. Yao is spoken in some parts of the country. Chichewa, the local language and is widely spoken in the whole Malawi.

With regard to internal validity, as outlined in lines 208 to 216, p.9. Thus, in the initial stage we engaged a multi-disciplinary team of local Malawian mental health experts. Two from the Northern region where Tumbuka is widely spoken, one from central where Chichewa is widely spoken and one from south but from a Yao speaking territory. They all had to translate the content and back translated. This was done to ensure accuracy and cultural sensitivity. The changes were approved via consensus. The draft consensus translation was then back translated into English and Chichewa by one independent bilingual Malawian with a background education in Social Sciences to check the original language for discrepancies and ensure the meaning is sustained. In addition, the preliminary testing of the adapted tool during focus group discussions and in-depth interviews ensured comprehensibility since this adapted tool is in both Chichewa the main local language and English.

3. Adaptation procedures

Reviewer comment: a. Lines 172 – 173: it would be helpful to state what the recommendations from the previous formative research were

Response: We thank you for your suggestion. We have included information on previous formative research we conducted (See 202 to 205, p.9).

Reviewer comment: 4. Curious to learn more about how the mental health experts were selected. In addition, how did the mental health clinicians categorized under ‘healthcare providers’?

Response: Thank you for your comment. The mental health experts were purposively selected based on their educational background and involvement in adolescents’ service delivery. These comprised of a Clinical psychologist, mental health nurse, psychosocial counsellor and psychiatric clinical officer who was also a Teen Club Coordinator. The mental health clinicians are clinical officers involved in ART program service delivery.

Reviewer comment: Were the mental health experts, healthcare providers, and key informants selected from the 2 ART centers too?

Response: The three mental health experts came from different workplaces (Clinical Psychologist is a Senior Lecturer and also Student Counsellor at the University of Malawi, Mental Health Nurse works for Zomba City Council Clinic, Psychosocial Counsellor works for Elizabeth Glaser Pediatric AIDS Foundation (EGPAF)) and the Psychiatric clinical officer works at one of the ART centers.

The health care providers were purposively selected from the 2 ART sites and the Key Informants came from the University of Malawi, EGPAF, District Health Office (DHO) and Zomba Central Mental Hospital

Reviewer comment: 6. What language were the interviews conducted in? It was stated that the focus groups were conducted in the local language but not clear what language the interviews were conducted in

Response: Thank you for your feedback. We have included a paragraph in lines 153 to 155, p.6 and 7 that “We conducted all FGDs in the local language (Chichewa). Some HCPs and KIs preferred to be interviewed in English, while others preferred or were comfortable to communicate in Chichewa”.

Reviewer comment: 7. The process of thematic analysis could be further expanded. How many people participated in the coding process, was inter-coder reliability, member checking or triangulation used? how was consensus reached? It might be helpful to add a thematic map.

Response: Thank you for your valuable suggestion. We have expanded the thematic analysis information. The primary author ECK did the entire data analysis. However, triangulation was used through using different data sources such as interview transcripts to see if they support similar themes and also by conducting interviews and focus group discussions. Additionally, we took notes during the FGDs, which complemented the audio recordings during the transcription process. A thematic map has been included in the manuscript (see lines 253 to 260, p. 11).

Results

Reviewer comment: 1. It would be good if the author presented a table with the original HEADSS, the adapted version and the final version.

Response: Thank you for your valuable suggestion. The original and adapted HEADSS have been included as Supporting Information (S3 Text. Original HEADSS tool and adaptation process notes and adapted versions of HEADSS are included as S4 Text. Participants HEADSS adaptation notes_v1, S5 Text. HEADSS adaptation v1, S6 Text. Participants HEADSS adaptation notes_ v2, S7 Text. HEADSS adaptation v2, S8 Text. HEADSS adaptation v3 and S9 Text. HEADSS adaptation _v4_Final Version as shown in lines 320 to 325, p. 15.

Reviewer comment: 2. Spelling errors in the results section—specifically the quote from the participants.

Response: Thank you for your valuable feedback. We have corrected the spelling errors in the quotes from participants throughout the results section as highlighted in yellow.

Reviewer comment 3. Line 480—the authors should clarify ‘expert client’. Also repeated in lines 553 – 554.

Response: Thank you. We have clarified in lines 346 to 351, p.16 that “Expert Clients are trained HIV Positive individual who workers in health facilities in Malawi due to shortage of health care professionals. They act as lay health care workers, peer educators and supporters and they provide counselling, psychosocial support and assistance in accessing HIV care to children as well as pregnant women”.

Discussion

Reviewer comment 1. Lines 617 – 618: citation needed

Response: We are very grateful for your suggestion. We have included citations (see lines 672 to 675, p.29).

Reviewer comment: 2. Future studies should include validation of the adapted HEADSS. Psychometric properties such as face, concurrent and convergent validity and reliability are essential to show the appropriateness of a screening tool.

Response: Thank you. We appreciate your suggestion. Currently, a validation study on this adapted HEADSS is in progress. However, we have also included a sentence on validation of adapted HEADSS as future recommendation (see lines 700 to 703, p.30).

Reviewer comment: 3. The authors should also touch on how the screening tool will be implemented and evaluated in clinical settings

Response: Thank you for your valuable suggestion. As mentioned above, we have included a sentence that currently a mixed-methods prospective quasi-experimental study employing a non-equivalent control group design and exploratory qualitative designs is being conducted in four sites to evaluate the HEADSS in clinical setting (see lines 700 to 703, p.30).

Reviewer comment:4. The authors should comment on how the elimination of suicide screening will affect the validity of the tool and the impact on mental health outcome. Suicide is rising amongst adolescent population and patients with HIV/AIDS are

Response: We appreciate your valuable suggestion. We have now included a paragraph on the importance of including suicide section in HEADSS assessment (see lines 675 to 685, p. 29 and 30).

Strengths

Reviewer comment:1. I wonder how the study setting provides strength for the study or how all the participants being on ARVs is a strength. Similarly, the authors should state why having participation of students in boarding school is a strength. What proportion of the participants are from boarding school?

Response: We believe that the study setting within the ART program is a strength, particularly since this adapted HEADSS tool focuses on ALHIV. Including ALHIV already enrolled in ART may further strengthen the study, as participants likely felt more comfortable in a familiar environment with their peer presence. Ethical considerations and concerns about non-disclosure prevented us from approaching ALHIV in the community. Although the sample of ALHIV in boarding schools was small (n=7), we considered it was important to include their voices and contributions, as they often face unique challenges and are frequently excluded from studies.

REVIEWER 2 COMMENTS

This is a very important and rigorously done work. Just a few comments:

Reviewer comment:1. It would have helped to have a table with a few example questions of the adapted tool perhaps juxtaposed with the relevant changed questions of the HEADDS version used in your earlier work

Response: We appreciate very much your suggestion. As mentioned earlier, we have now included the original and adapted HEADSS have been included as Supporting Information (S3 Text. Original HEADSS tool, S4 Text. HEADSS adaptation v1, S5 Text. HEADSS adaptation v2, S6 Text. HEADSS adaptation v3, S7 Text. HEADSS adaptation _Final Version as shown in lines 321 to 323, p. 15 and also in lines 737 to 743, p.32.

Reviewer comment:2. Line 79: Although referenced it will help to briefly outline here the findings in your earleir work that suggested cultural modifications need to be done.

Response: Thank you for your suggestion. We have included some information of prior findings of our formative study (see lines 202 to 205, p. 9).

A few edits:

Reviewer comment :3. Line 435: , remove the word "to" before Malawi to make the sentence easier to understand

Response: We appreciate your suggestion. We have removed “to” in the sentence.

Reviewer comment:4. Line 556: add help before ALHIV

Response: Thank you for your observation. We have added the word “help” before ALHIV (see line 611, p. 26).

Reviewer comment :5. Line 593: the sentence "HEADSS screening has in the identification of psychosocial issues, such as depression and high-risk behaviors in adolescents (20,21,24,25,28–31,55)".is hanging and not clear.

Response: We appreciate your feedback. We have amended the sentence and it reads “HEADSS screening plays a role in the identification of psychosocial issues, such as depression and high-risk behaviors, among adolescents”. (See lines 646 to 648, p.28).

Reviewer comment:6. Line 661-664: This last sentence under conclusion about training could be left out without loss to the paper as it does not tie in with the overall aim of the study and the discussion

Response: Thank you so much for your suggestion. We have removed the last sentence about training.

---

## [Decision Letter · Decision Letter 1]

4 Jun 2025

Dear Dr. Kip,

Thank you for submitting your manuscript to PLOS ONE. After careful consideration, we feel that it has merit but does not fully meet PLOS ONE’s publication criteria as it currently stands. Therefore, we invite you to submit a revised version of the manuscript that addresses the points raised during the review process.

**ACADEMIC EDITOR:** Authors should address the comments recommended by Reviewer I, especially considering removing unnecessary repetition. Please include the reason for using a purposive sampling method and highlight the justification for your sample size. Could the sampling method be a limitation of this study? Also, kindly make the abstract section brief but comprehensive.

We look forward to receiving your revised manuscript.

Kind regards,

Anthony A. Olashore, MBCHB, PhD.

Academic Editor

PLOS ONE

Journal Requirements:

Reviewers' comments:

Reviewer's Responses to Questions

**Comments to the Author**

Reviewer #1: (No Response)

Reviewer #2: All comments have been addressed

2. Is the manuscript technically sound, and do the data support the conclusions?

Reviewer #1: Yes

Reviewer #2: Yes

3. Has the statistical analysis been performed appropriately and rigorously?

Reviewer #1: Yes

Reviewer #2: Yes

4. Have the authors made all data underlying the findings in their manuscript fully available?

Reviewer #1: Yes

Reviewer #2: Yes

5. Is the manuscript presented in an intelligible fashion and written in standard English?

Reviewer #1: Yes

Reviewer #2: Yes

Reviewer #1: Thank you for asking me to review the revised manuscript. The revised version is much improved. I have a few suggestions for further revision:

1. Lines 253–260: I recommend deleting this section, as it reiterates points already made earlier in the manuscript.

2. Lines 321–325: Please revise the reference to the supplemental materials to read: “see Supplemental Information S3–S9.”

3. Lines 337–339: This section needs clarification. Elsewhere in the manuscript, it appears that suicide-related questions were included. Was there a period during which these questions were removed? This does not seem to be the case based on the supplemental materials.

4. Lines 337–339: I also recommend adding a comment—perhaps in the Discussion—on the implications of removing sexual orientation questions. Consider addressing how stigma and psychological distress among sexual minorities in Malawi may influence HIV transmission, treatment adherence, risky sexual behaviors, and increased risk of mental health disorders among sexual minority individuals living with HIV.

5. Lines 382–387: The addition of a section on suicide in the Discussion is valuable. I suggest expanding this section, particularly by contrasting participants’ generally supportive views on asking about suicide with healthcare workers’ concerns that such questions may heighten suicide risk. This dichotomy is worth further exploration—specifically, how prevalent this attitude gap is in the existing literature. The authors should also consider discussing how this divergence in perspectives may impact suicide screening and prevention efforts, and how it highlights the need for interventions—such as provider training and public health messaging—to shift healthcare workers’ attitudes toward alignment with evidence-based best practices, especially for adolescents living with HIV.

6. The manuscript contains multiple sections that repeat similar ideas (e.g., discussion of acceptability, relevance, and comprehensibility is revisited with overlapping quotes and interpretations). Consider condensing.

Reviewer #2: (No Response)

**Do you want your identity to be public for this peer review?** For information about this choice, including consent withdrawal, please see our Privacy Policy

Reviewer #1: **Yes: ** Temitope Ogundare, MD, MPH

Reviewer #2: No

---

## [Author Response · Author response to Decision Letter 2]

16 Jul 2025

Authors Response to Reviewers

P. O Box 212

Zomba

Malawi

15th July, 2025

To:

Academic Editor

PLOS ONE

Dear Academic Editor Prof Olashore,

Ref: Re-submission ID PONE-D-24-58777R1: Cultural adaptation of a psychosocial screening tool for adolescents living with HIV/AIDS attending antiretroviral therapy program in Malawi

Authors:

Esther C. Kip (esther_kip@yahoo.com)

Mina C. Hosseinipour (mina_hosseinipour@med.unc.edu)

Kazione Kulisewa (kkulisewa@kuhes.ac.mw)

Brian W. Pence (bpence@unc.edu)

Michael Udedi (mphatsoudedi@yahoo.co.uk)

Bradley N. Gaynes (bgaynes@med.unc.edu)

Vivian F. Go (vgo@live.unc.edu)

Version 2: Date:15/07/2025

To the Academic Editor:

Once again, we are very grateful to the editors and reviewers for their positive and constructive suggestions. We feel that the quality of the revised manuscript has been improved significantly as a result. All comments in the manuscript file have been taken into account. We have followed each of the Editor’s recommendations for the revised submission, including highlighting in yellow our modifications

ACADEMIC EDITOR

Please include the reason for using a purposive sampling method and highlight the justification for your sample size

Response: Thank you for your suggestion. We have included the rationale for using purposive sampling method and highlighted the justification for the sample size in the manuscript as shown in lines 141 - 158, p. 6 and 7

Could the sampling method be a limitation of this study?

Response: Thank you for pointing out this issue. Yes, it might be a limitation of this study due to limited generalizability of the findings and selection bias of participants. We have included this in the Study Limitation section (see lines 711 – 715, pgs. 32 & 33).

Also, kindly make the abstract section brief but comprehensive

Response: Thank you for your valuable feedback. We have revised the abstract section to be more concise, reducing its length from 299 words to 242 words (see lines 24 - 50, pgs. 1&2).

Response: We appreciate this important point. Upon thorough review, no explicit evidence of retracted articles was found among the cited references but we have revised reference No.55 to read “Government of the Republic of Malawi. National Commission for Science and Technology. National policy measures and requirements for the improvement of health research co-ordination in Malawi. 2012 revised edition. Lilongwe. Malawi”. We have also highlighted it in the reference list. However, we acknowledge the possibility of oversight. We would be grateful if the editors could indicate any specific retracted articles they have identified.

REVIEWER 1 COMMENTS

Reviewer comment: 1. Lines 253–260: I recommend deleting this section, as it reiterates points already made earlier in the manuscript.

Response: We thank you for your suggestion. The information in lines 253 -260 has been deleted.

Reviewer comment: 2. Lines 321–325: Please revise the reference to the supplemental materials to read: “see Supplemental Information S3–S9.”

Response: Thank you for your valuable feedback. We have revised the reference to the supplemental materials to read as suggested (see lines 325 – 326, p.16).

Reviewer comment: 3. Lines 337–339: This section needs clarification. Elsewhere in the manuscript, it appears that suicide-related questions were included. Was there a period during which these questions were removed? This does not seem to be the case based on the supplemental materials.

Response: Thank for your feedback. As recommended, we have included some clarification on the questions which were removed/changed from the original HEADSS tool during the adaptation process as shown in lines 337 – 347, p. 15 &16.

Reviewer comment: 4. Lines 337–339: I also recommend adding a comment—perhaps in the Discussion—on the implications of removing sexual orientation questions. Consider addressing how stigma and psychological distress among sexual minorities in Malawi may influence HIV transmission, treatment adherence, risky sexual behaviors, and increased risk of mental health disorders among sexual minority individuals living with HIV.

Response: Thank you for pointing this out. We have included the suggested information in the Discussion section as highlighted in yellow in lines 610-638, pgs. 28– 29.

Reviewer comment: 5. Lines 382–387: The addition of a section on suicide in the Discussion is valuable. I suggest expanding this section, particularly by contrasting participants’ generally supportive views on asking about suicide with healthcare workers’ concerns that such questions may heighten suicide risk. This dichotomy is worth further exploration—specifically, how prevalent this attitude gap is in the existing literature. The authors should also consider discussing how this divergence in perspectives may impact suicide screening and prevention efforts, and how it highlights the need for interventions—such as provider training and public health messaging—to shift healthcare workers’ attitudes toward alignment with evidence-based best practices, especially for adolescents living with HIV.

Response: Thank you again for your valuable suggestion. We have included the suggested information in the Discussion section as highlighted in yellow (see lines 645 – 651 & 660 – 674, pgs. 30 & 31).

Reviewer comment: 6. The manuscript contains multiple sections that repeat similar ideas (e.g., discussion of acceptability, relevance, and comprehensibility is revisited with overlapping quotes and interpretations). Consider condensing.

Response: We appreciate your suggestion. We have incorporated your suggestions in the results section as shown in line 328, we have combined the section on Comprehensibility and Relevance of the adapted tool and all section and subsections in Acceptability section have been combined as shown in line 421.

---

## [Decision Letter · Decision Letter 2]

1 Aug 2025

Cultural adaptation of a psychosocial screening tool for adolescents living with HIV/AIDS attending antiretroviral therapy program in Malawi

PONE-D-24-58777R2

Dear Dr. Kip,

We’re pleased to inform you that your manuscript has been judged scientifically suitable for publication and will be formally accepted for publication once it meets all outstanding technical requirements.

Kind regards,

Anthony A. Olashore, MBCHB, PhD.

Academic Editor

PLOS ONE

Additional Editor Comments (optional):

Reviewers' comments:

Reviewer's Responses to Questions

**Comments to the Author**

Reviewer #1: (No Response)

2. Is the manuscript technically sound, and do the data support the conclusions?

Reviewer #1: Yes

3. Has the statistical analysis been performed appropriately and rigorously?

Reviewer #1: Yes

4. Have the authors made all data underlying the findings in their manuscript fully available?

Reviewer #1: Yes

5. Is the manuscript presented in an intelligible fashion and written in standard English?

Reviewer #1: Yes

Reviewer #1: Thank you for asking me to review this revised manuscript. The authors have addressed all comments and have implemented them.

**Do you want your identity to be public for this peer review?** For information about this choice, including consent withdrawal, please see our Privacy Policy

Reviewer #1: **Yes: ** Temitope Ogundare, MD, MPH

---

## [Editor Report · Acceptance letter]

PONE-D-24-58777R2

PLOS ONE

Dear Dr. Kip,

I'm pleased to inform you that your manuscript has been deemed suitable for publication in PLOS ONE. Congratulations! Your manuscript is now being handed over to our production team.

Kind regards,

on behalf of

Dr. Anthony A. Olashore

Academic Editor

PLOS ONE